# Cross-Scale MAE: A Tale of Multi-Scale Exploitation in Remote Sensing

**Maofeng Tang[1], Andrei Cozma[1], Konstantinos Georgiou[1], Hairong Qi[1]**
[1]Min H. Kao Department of Electrical Engineering and Computer Science
University of Tennessee, Knoxville
{mtang4, acozma, kgeorgio}@vols.utk.edu, hqi@utk.edu,

## Abstract

Remote sensing images present unique challenges to image analysis due to the extensive geographic coverage, hardware limitations, and misaligned multi-scale images. This paper revisits the classical multi-scale representation learning problem but under the general framework of self-supervised learning for remote sensing image understanding. We present Cross-Scale MAE, a self-supervised model built upon the Masked Auto-Encoder (MAE). During pre-training, Cross-Scale MAE employs scale augmentation techniques and enforces cross-scale consistency constraints through both contrastive and generative losses to ensure consistent and meaningful representations well-suited for a wide range of downstream tasks. Further, our implementation leverages the xFormers library to accelerate network pre-training on a single GPU while maintaining the quality of learned representations. Experimental evaluations demonstrate that Cross-Scale MAE exhibits superior performance compared to standard MAE and other state-of-the-art remote sensing MAE methods.

## 1 Introduction

Remote sensing image understanding has been reaping the benefits of recent advances in computer vision while at the same time posing unique challenges. First, remote sensing imagery usually covers the Earth's surface extensively for Earth observation purposes [52, 43, 31]. Compared to the vast area the images cover, even a large amount of training samples would become sparse. Furthermore, generating a set of representative training samples is challenging, especially in less-explored regions. Hence, self-supervised learning (SSL) becomes a viable solution since no training data is required for representation learning purposes. A recently published paper [46] provides a comprehensive review of SSL in remote sensing.

The second challenge arises from the inherent hardware limitations of remote sensing devices, where one can only expect to acquire images of high resolution in the spatial, spectral, or temporal domain, but not all. These devices usually sacrifice spatial resolution to gain spectral and/or temporal resolution for different functional and/or material analyses of the Earth's surface. Ground Sample Distance (GSD) measures the spatial resolution, denoting the physical distance between two adjacent pixels. For example, Landsat-8 [45] has 8 spectral bands at 30 m GSD; Sentinel-2 [17] has 4 bands at 10 m, 6 bands at 20 m, and 3 bands at 60 m GSD; and WorldView-3 [29] has 8 multi-spectral bands at 1.24 m GSD. These multi-scale images, although imaging the same site, might not be aligned at all, and the alignment process can be both time-consuming and expensive.

It is worth noting that in remote sensing imagery, due to the rich and unique data provided along the temporal and spectral domains, algorithms tend to exploit more on the spectral and temporal organization but neglect the spatial characteristics, as pointed out in [4]. So, extracting effective representations from these misaligned multi-scale images poses a significant challenge. In this

37th Conference on Neural Information Processing Systems (NeurIPS 2023).

paper, we delve into multi-scale analysis and develop novel ways to fully explore the wealth of information offered by images at multiple scales. We present Cross-Scale MAE, a self-supervised model based on the Masked Autoencoder (MAE) [21] that explicitly learns relationships between data at different scales throughout the pre-training process. By leveraging this information, Cross-Scale MAE produces a robust pre-trained model with superior performance across various GSDs and tasks.

Like MAE, Cross-Scale MAE also masks most of the transformed image and then tasks a Vision Transformer (ViT)-based Autoencoder with embeddings of the unmasked components. A decoding light ViT then decodes the entire image from these learned embeddings, where the decoder is later discarded, and the encoder is used to produce representations for an unmasked input image. The main differences can be addressed from two aspects. First, Cross-Scale MAE introduces the "scale augmentation" to synthesize inputs of different scales of one input image to engage the transformer to learn features with varying scales in the encoder. The scale augmentation eliminates the need for aligned multi-scale images of the same ground site as input. The pre-trained model is generic enough to extract arbitrary scale input image representations. The second key difference stems from the term "Cross-Scale", where features extracted from the multi-scale images are not simply concatenated or combined in any ad hoc fashion as many existing fusion frameworks do; instead, Cross-Scale MAE exploits information consistency across the multi-scale images and uses it as constraints to train a more robust Autoencoder model. The cross-scale information consistency is examined at both the encoder and decoder end to ensure consistency is reinforced at both the structural level (i.e., the encoder end) and the semantic level (i.e., the decoder end).

The experimental results overwhelmingly show that Cross-Scale MAE yields more robust multi-scale representations and better performance for downstream tasks as compared to other state-of-the-art methods, across a couple of remote sensing datasets with a variety of scale and resolution characteristics. Our contributions can be summarized as follows:

(1) We develop Cross-Scale MAE, a flexible SSL framework that yields robust representations by enforcing cross-scale information consistency at both structural and semantic levels without the need for aligned multi-scale remote sensing imagery.

(2) We investigate the combination of contrastive learning and masked imagery modeling, specifically, the effect of negative samples on representation at different levels.

(3) We deploy xFormers to realize Cross-Scale MAE, where both the pre-training time and memory usage are improved without performance degradation, making large language model training affordable on a single GPU.

The remainder of this paper is organized as follows. Sec. 2 reviews recent developments in SSL and multi-scale representation learning. Sec. 3 elaborates on the proposed Cross-Scale MAE model design. Sec. 4 presents details about experiments and results. Sec. 5 concludes the paper and provides general directions for further improvements.

## 2   Related Work

We mainly discuss recent works related to SSL and multi-scale representation learning.

**Self-Supervised Learning (SSL).** Self-supervised learning is a branch of unsupervised learning aiming to learn effective representations for downstream tasks without using training labels. SSL can be divided into two groups, namely, discriminative models and generative models. The discriminative approach, e.g., contrastive learning (CL), requires the representation to be distinctive in information for different inputs. The gist of CL is to make the representations of positive samples close and those of negative samples far from each other, where positive samples are obtained by applying various augmentation schemes on the same image. CL has emerged as a dominant visual SSL method, especially after the advent of MoCo [20] and SimCLR [7]. Negative-free, i.e., non-contrastive, joint-embedding methods have been developed [51, 50, 9], demonstrating comparable performance as CL methods. In the remote sensing community, the CL-based representation learning methods have been developed, such as SeCo [28], mCL-LC [42], where the SeCo, short for Seasonal Contrast, offers a novel approach to the challenge of insufficient labeled data in remote sensing by capitalizing on self-supervised learning mechanisms. The mCL-LC method jointly learns global-local representations to improve the segmentation performance for satellite images. On the other hand, the generative approach, e.g., MAE, requires the representation to preserve as much of the input information as

possible. MAE is an autoencoder with masked prediction, i.e., predicting a property of masked input from unmasked input. Benefiting from the success of masked language models [34, 35, 1, 13], the similar mechanism was investigated in vision tasks and has dramatically advanced the field of SSL [16, 21]. Following the trend in computer vision, the masked image model has also shown strength in the remote sensing field. For example, the GFM model [30] excels in representation learning through continual learning, aiming to enhance the applicability of large language models to satellite imagery via knowledge distillation.

In Cross-Scale MAE, both the discriminative and generative mechanisms are used to guarantee the representation from images of different scales to be consistent and meaningful.

**Multi-Scale Representation Learning.** As mentioned in Sec. 1, the multi-scale phenomenon is common in vision tasks. To leverage the information in multi-scale sources, the vision community has proposed to extract multi-scale features using traditional approaches, including, for example, spatial pyramids [39, 2, 26], dense sampling of windows [23, 48, 49], and the combination of them [19]. In the last decade, deep convolution neural networks (CNNs) have emerged as the de facto architecture for a wide range of vision tasks. Because of the pooling and multi-convolution kernel operation, the CNN constructs the feature pyramids inherently so that it has been used to build deep multi-scale representations [24]. CNN-based multi-scale representation learning is commonly approached in two ways. The first approach involves utilizing external or preset factors, which can include multi-scale kernel architecture [41, 25, 37, 53] and multi-scale inputs architecture [14, 18, 27]. The second approach involves designing internal layers of the network, such as skip and dense connections [40, 33, 5]. Over the past three years, there has been a notable surge of interest in applying transformer-based architectures to computer vision tasks. Among these architectures, the Vision Transformer (ViT) [16] stands out as a particularly successful example, where, compared to CNN, ViT balances the global and local features much better. Recently, other efforts have been made for the multi-scale feature learning, including the multi-scale Deformable Attention and Multi-level Features Aggregation (MSDAM and MLFAM) network [15], and the Shunted Self-Attention (SSA) network [38]. These methods have shown strength in multi-level feature extraction for the natural image containing multiple objects of different sizes. However, in remote sensing images, the multi-scale phenomenon is even more prevalent and dynamic. As a result, there has been a growing recognition of the need for customized approaches to address this. The recently published Scale-MAE [36] is one such example.

Essentially, compared with the most recent two state-of-the-art MAE-based satellite imagery representation learning methods, i.e., SatMAE [12], and Scale-MAE [36], SatMAE is the first paper that applies MAE to extract representations from satellite images with single and fixed scale. Scale-MAE and the proposed Cross-Scale MAE are based on MAE but focus on multi-scale characteristics. Specifically, Scale-MAE develops the GSD position encoding and applies multi-de-convolution after the decoder to reconstruct images of different scales. Nonetheless, Scale-MAE integrates the scale information into the network via hard coding of known GSD, and the de-convolution can only result in a specific scale ratio. The proposed Cross-Scale MAE designs the network to learn the information across different scales. With scale augmentation and multi-level contrastive loss between the scale pair and masked patches reconstruction, Cross-Scale MAE can learn informative and consistent representation across different scales without the need of known GSD. Additionally, we leverage xFormers on a single GPU for pre-training efficiency.

## 3 Methodology

This section elaborates on the Cross-Scale MAE pre-training framework as illustrated in Fig. 1. Cross-Scale MAE is a self-supervised pre-training framework built upon the MAE [21]. Its objective is to learn consistent representations from multi-scale images captured at the same site by harnessing the strengths of both discriminative and generative learning approaches. Cross-Scale MAE incorporates two novel loss functions at the encoder and decoder stages to achieve this objective. Firstly, the representations obtained from images of different scales are enforced to be consistent. This ensures that the learned representations capture relevant information *across* scales. Secondly, a decoder is employed to reconstruct the corresponding input, promoting the learned representation to be more representative. Additionally, the decoder utilizes cross-prediction, where the embeddings generated by the decoder from an image with a lower GSD are used to predict the embeddings from an image

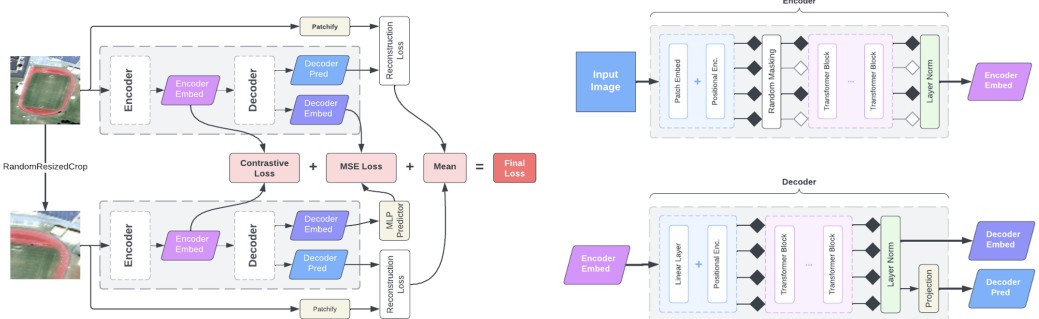

Figure 1: The architecture of Cross-Scale MAE comprises an encoder and a decoder (Left). The encoder (Top-Right) employs a vision transformer (ViT) backbone, specifically a ViT-base with 12 self-attention blocks. The decoder (Bottom-Right) uses a lightweight ViT backbone with 8 self-attention blocks. A single satellite image undergoes scale augmentation through random cropping and resizing. The contrastive loss is computed using encoder outputs for the two scale inputs. The cross-prediction loss (in MSE) is applied to the last self-attention block's decoder output. Reconstruction loss compares the predicted masked patch and the actual input.

with a higher GSD. This cross-prediction mechanism helps capturing and leveraging the multi-scale information present in the images.

## 3.1 Basic Setting

Let $p \in \mathbb{R}^{H \times W \times C}$ represent an input image of height $H$ and width $W$ with $C$ channels. First, multi-scale augmentation (Sec. 3.2) is applied to $p$ to generate two images, $p_h$ of relatively higher GSD and $p_l$ of relatively lower GSD. For each of the augmented images, a sequence of $|S|$ independent patches of height and width of $n$ pixels are created, where each of the patches, $s \in S$, has dimension $s \in \mathbb{R}^{n^2 C}$. A fraction, $m$, of the patches is removed, while the remaining patches are passed through a projection function (e.g., a linear layer) to project the patches $s$ into $D$ dimensions, $f_{emd} : \mathbb{R}^{n^2 C} \to \mathbb{R}^D$, to obtain embedded patches $S_{emd} = f_{emd}(S)$. A standard positional encoding vector is added to the embedded patches and fed into the encoder. After the encoder, the removed $m$ patches are placed back into their original location in the sequence of patches where a learned mask token represents the masked patches that were not encoded. Another positional encoding vector is added to all patches, and a sequence of transformer blocks decodes these patches to form the original input image; these steps are illustrated in Fig. 1.

## 3.2 Multi-Scale Augmentation

A specialized augmentation method is applied to the input image before patchifying it in the Cross-Scale MAE framework. This augmentation process generates an additional image of lower GSD for the same site, and the two images are then used as inputs through two independent instances of the MAE network. More specifically, given an input image $p$ with a certain GSD, the framework randomly selects two scale ratios, $r_h$ and $r_l$, from the range [0, 1], where $r_l < r_h$. The image $p$ is then down-sampled using these scale ratios to generate two images, $p_h$ and $p_l$. In our experiments, we set $r_h = 1$, so $p_h = p$. Both images, $p_h$ and $p_l$, are subsequently resized to a fixed size of $128\text{px}^2$, and the GSD values of $p_h$ and $p_l$ are denoted as $g_h$ and $g_l$, respectively.

Note that this augmentation approach differs from traditional augmentation, where the rescaled images are traditionally only used as separate samples. Instead, the multi-scale inputs are generated dynamically at runtime and used as inputs through multiple network branches, where the branches have shared weights. The constraints enforced upon the learned representations of these network branches are detailed in the following sections.

### 3.3 Cross-Scale Information Consistency in the Encoder

Typically, the standard MAE trains a network to reconstruct an image by masking out a significant portion of its pixels. However, the standard MAE encoder does not explicitly address the issue of representation consistency for multi-scale input sources. In contrast, the proposed Cross-Scale MAE aims to learn meaningful and consistent scale-invariant representations. To achieve this, Cross-Scale MAE focuses on two fundamental properties in the learned representations: (1) consistency of information across different-scale images and (2) representativeness of the raw image. In the Cross-Scale MAE framework, a contrastive learning strategy is employed within the encoder to ensure consistency among representations derived from different-scale images.

The intuitive idea behind Cross-Scale MAE is that different-scale images captured at the same ground site contain highly relevant information despite potential visual differences. Cross-Scale MAE leverages this notion by maximizing the shared information between representations derived from different-scale images of the same site while minimizing the shared information between images from different sites. This approach aligns with the principles of contrastive learning. In contrastive learning, the InfoNCE loss is commonly used to estimate the lower bound of mutual information between data samples. By minimizing the InfoNCE loss, we effectively maximize the mutual information between representations from different scales [44, 32].

Specifically, the encoder $E$, uses the ViT-base [16] as the backbone. Upon feeding an input image $p$ to encoder $E$, a high-level representation, $E(p)$, can be obtained. Before computing the contrastive loss, a nonlinear projection head $g_f(\cdot)$ is needed, which has been proven to be effective [8]. So, after the encoder and projection head, we obtain the representation, $z = g_f(E(p))$. The cross-scale consistency loss is thus defined as:

$$\mathcal{L}_{cc} = \frac{1}{2N} \sum_{k=1}^{N} \left( \ell_{info}\left(p_l^k, p_h^k\right) + \ell_{info}\left(p_h^k, p_l^k\right) \right) \tag{1}$$

where $p_h^k$ and $p_l^k$ are the different-scale images from the same input image $p_k$, $N$ denotes the number of samples in a mini-batch, and the InfoNCE contrastive loss function $\ell_{info}$ is the same as in [6], which is defined as follows:

$$\ell_{info}\left(p_l^k, p_h^k\right) = -\log \frac{\exp\left(\text{sim}\left(z_l^k, z_h^k\right)/\tau\right)}{\sum_{p \in \Lambda} \exp\left(\frac{\text{sim}\left(z_l^k, g_f(E(p))\right)}{\tau}\right)} \tag{2}$$

where $z_l^k = g_f\left(E\left(p_l^k\right)\right)$, $z_h^k = g_f\left(E\left(p_h^k\right)\right)$ and $\Lambda$ is the sample batch. The sim operator is the cosine distance.

### 3.4 Cross-Scale Prediction in the Decoder

The InfoNCE loss in the encoder provides consistency for multi-scale input but cannot guarantee that the learned representation adequately represents the input. In other words, the representation learned by the network from multi-scale images should be not only consistent among different scales, but also representative of the semantic information. According to [3], the self-supervised ViT automatically learns class-specific features leading to unsupervised object segmentation; specifically, the attention block's last layer includes abundant semantic information. Motivated by [3], the decoder of the Cross-Scale MAE has two tasks: (1) reconstruction of the multi-scale input as close as possible, (2) cross-scale prediction between attentions from different scales, as shown in the decoder part of Fig. 1 and detailed below.

Similar to the standard MAE, Cross-Scale MAE also adopts a light decoder. Decoding follows the standard MAE decoder where, following the encoder, the removed $m$ patches are placed back into their original location in the sequence of patches where a learned mask token represents the masked patches that were not encoded, and a positional encoding is added. Then, a series of transformer layers decode all patches. Given the two representations $f_{e,h}$ and $f_{e,l}$, where $f_{e,h} = E(p_h)$, $f_{e,l} = E(p_l)$, after the decoder attention block, we will obtain two new representations, $f_{d,h}$ and $f_{d,l}$. Then, the cross-scale prediction is applied between them to get the cross-scale prediction loss, defined as:

Table 1: The information of datasets used in Sec. 4

| Dataset | GSD(m) | Number of Images | Number of Categories |
|---------|--------|------------------|----------------------|
| RESISC45 | 0.2-30 | 31500 | 45 |
| WHU-19 | 0.5 | 1050 | 19 |
| UC Merced | 0.3 | 2100 | 21 |
| EuroSAT | 10 | 27000 | 10 |

$$\mathcal{L}_{cp} = \frac{1}{N} \sum_{k=1}^{N} \left( \ell_{pred} \left( f_{d,l}^k, f_{d,h}^k \right) \right) \tag{3}$$

where $N$ is the batch size and the prediction loss $\ell_{pred}$ is defined as:

$$\ell_{pred} \left( f_{d,l}^k, f_{d,h}^k \right) = ||f_{d,h}^k - g_p(f_{d,l}^k)||_2^2 \tag{4}$$

It follows the Mean Squared Error (MSE) between the prediction and target representation, where the predictor, $g_p(\cdot)$, is a multi-layer perceptron (MLP). Besides the cross-scale prediction loss, the decoder reconstructs images of different scales. The reconstruction loss is thus defined as:

$$\mathcal{L}_{re} = \frac{1}{N} \sum_{k=1}^{N} \left( ||p_l - \tilde{p}_l||_2^2 + ||p_h - \tilde{p}_h||_2^2 \right) \tag{5}$$

where $\tilde{p}_l = D(E(p_l))$ and $\tilde{p}_h = D(E(p_h))$. Via the cross-scale prediction loss and the reconstruction loss, the consistency and effectiveness of the semantic information of the representation can be better preserved.

The total loss of the Cross-Scale MAE is the sum of cross-scale consistency loss at the encoder ($\mathcal{L}_{cc}$), cross-scale prediction loss at the decoder ($\mathcal{L}_{cp}$), and the reconstruction loss ($\mathcal{L}_{re}$), defined as:

$$\mathcal{L} = \mathcal{L}_{cc} + \mathcal{L}_{cp} + \mathcal{L}_{re} \tag{6}$$

## 4 Experiments & Results

We investigate the quality of representations learned from Cross-Scale MAE pre-training through comparisons to state-of-the-art approaches (Sec. 4.1) and comprehensive ablation studies (Sec. 4.2). Sec. 4.3, we briefly explore their robustness to scale variation and transfer performance to additional tasks. We use SatMAE [12] as the baseline, the state-of-the-art MAE-based approach for remote sensing image analysis. We pre-train Cross-Scale MAE with a ViT-Large model (unless mentioned otherwise) using the Functional Map of the World (fMoW) [11] RGB training set, which consists of 363.6k images of varying image resolution and GSD. In Sec. 4.1, we present experiments and results to validate the effectiveness of the proposed method, compared with baseline and other state-of-art methods, such as Scale-MAE [36]. Section 4.2 presents a set of ablation experiments to show how different losses and hyper-parameters affect the performance of Cross-Scale MAE. Section 4.3 briefly describes the efficient backbone built through xFormers.

### 4.1 Comparison with State-of-the-Art

Similar to Scale-MAE, we use both the K-nearest neighbors (KNN) classification accuracy and performance in downstream tasks as metrics to evaluate the quality of the representation learned by Cross-Scale MAE. Specifically, we pre-train the Cross-Scale MAE on the fMoW-RGB dataset and freeze the encoder as a representation generator; then, we test its effectiveness on different datasets via KNN classification.

**KNN Performance.** Similar to the original MAE configuration, the encoder of Cross-Scale MAE uses a ViT-Large model, and the decoder is a light ViT model with the 8-layer attention blocks. The datasets we use include RESISC45 [10], WHU-RS19 [47], UC Merced [47], EuroSAT [22], as shown in Table 1. To evaluate the representation learning capacity of the proposed method with

Table 2: Average KNN accuracy with different scale ratios ($12.5\%, 25\%, 50\%, 100\%$)

|  | RESISC45 | WHU-RS19 | UC Merced | EuroSAT |
|---|---|---|---|---|
| SatMAE | 66.3 | 69.9 | 69.7 | 81.9 |
| Scale-MAE | 70.0 | 79.5 | 75.0 | 86.7 |
| Cross-Scale MAE | 75.6 | 79.8 | 74.5 | 87.8 |

different GSD images, we evaluate the network performance using images with different scale ratios, $\{12.5\%, 25\%, 50\%, 100\%\}$, to the raw image to simulate different GSD.

The results are presented in Fig. 2. This figure shows the performance of Cross-Scale MAE compared to both SatMAE and Scale-MAE for different scale ratios. From Fig. 2, we observe that, in general, the Cross-Scale MAE performs better than SatMAE or Scale-MAE in all different scale ratios on all datasets. And, for scale ratio in $\{25\%, 50\%, 100\%\}$, the Cross-Scale MAE can obtain more stable performance compared with SatMAE or Scale-MAE. When the scale ratio equals $12.5\%$, all three models perform relatively worse. This is because the scale ratio set during pre-training is from 0.2 to 0.8, and 0.125 is out of this range. Nonetheless, Cross-Scale MAE still presents overwhelmingly better performance than the other two. Table 2 shows the average accuracy at all scale ratios and compares Cross-Scale MAE with SatMAE and Scale-MAE. We can find that Cross-Scale MAE presents overwhelmingly better performance than SatMAE in all datasets and we observe similar trend as in Fig. 2. Cross-Scale MAE generally performs better than Scale-MAE except in the UC Merced dataset. It may be because RESISC45 covers an extensive range of GSD, from $0.2m$ to $30m$, but the other three datasets have fixed GSD. This comparison effectively demonstrates the robustness of representations generated from Cross-Scale MAE, especially on multi-scale datasets.

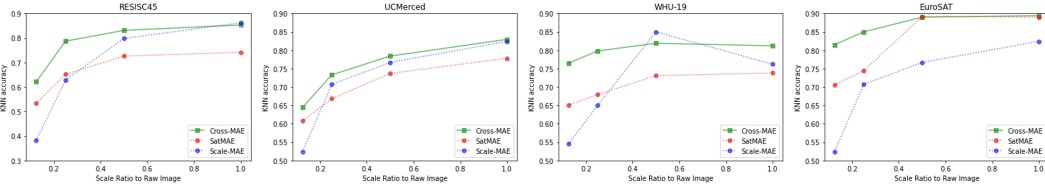

Figure 2: The KNN classification of Cross-Scale MAE for different datasets.

**Downstream Tasks.** We further evaluate the performance of Cross-Scale MAE on different downstream tasks to assess the effectiveness of the learned representations in various practical scenarios, including classification and segmentation. We conduct the classification task on the fMoW-RGB dataset. The segmentation downstream tasks are performed on two datasets, Potsdam and Vaihungen. We fine-tune the model for 50 epochs for all downstream tasks, following the same hyper-parameter settings as Scale-MAE [36].

To test the capability of handling multi-scale inputs, we compare the performance of different methods on low-resolution images with different scaling ratios applied on full-resolution images. In Table 3, we report the classification performance regarding Top-1 and Top-5 accuracy. In Table 4, we report the segmentation performance regarding mIoU. The proposed Cross-Scale MAE performs superior to state-of-the-art methods in both classification and semantic segmentation downstream tasks.

## 4.2 Ablation Study

The ablation study is four-fold. First, we investigate the critical role of each of the three loss functions to multi-scale inputs. Second, we investigate the effect of negative samples in different levels of the representation. This includes two parts, with the first evaluating the whole Cross-Scale MAE framework and the second under the pure contrastive learning framework (leaving out the masking patch reconstruction part). Third, we compare the effect of multi-scale input and the GSD positional encoding used in Scale-MAE [36]. Although we have shown performance improvement with scale augmentation, we extend the investigation to see whether it has better generalization capacity than

Table 3: Linear classification fine-tuning performance on fMoW-RGB

|  | | Scale=50% | | Scale=100% | |
|---|---|---|---|---|---|
|  | Backbone | Top-1 | Top-5 | Top-1 | Top-5 |
| GASSL | ResNet50 | 0.494 | 0.540 | 0.683 | 0.715 |
| SeCo | ResNet50 | 0.508 | 0.602 | 0.614 | 0.796 |
| SatMAE | ViT-Base | 0.551 | 0.776 | 0.651 | 0.892 |
| SatMAE | ViT-Large | 0.591 | 0.793 | 0.678 | 0.923 |
| Scale-MAE | ViT-Base | 0.623 | 0.850 | 0.673 | 0.930 |
| Cross-Scale MAE | ViT-Base | **0.632** | **0.914** | **0.692** | **0.954** |

Table 4: Semantic segmentation performance on Potsdam & Vaihingen (mIoU)

|  | | Potsdam | | | Vaihingen | | |
|---|---|---|---|---|---|---|---|
|  | Backbone | 25% | 50% | 100% | 25% | 50% | 100% |
| Supervised Baseline | ResNet50 | 0.4518 | 0.5212 | 0.7411 | 0.4603 | 0.5237 | 0.7438 |
| SimCLR | ResNet50 | 0.5327 | 0.5472 | 0.7233 | 0.5286 | 0.5904 | 0.7075 |
| BYOL | ResNet50 | 0.5509 | 0.5715 | 0.7264 | 0.5618 | 0.5903 | 0.7298 |
| SimSiam | ResNet50 | 0.5439 | 0.5903 | 0.7188 | 0.5522 | 0.6005 | 0.7205 |
| SeCo | ResNet50 | 0.5571 | 0.5935 | 0.7298 | 0.5741 | 0.6213 | 0.7250 |
| mCL-LC | ResNet50 | 0.5589 | 0.5973 | 0.7316 | 0.5795 | 0.6277 | 0.7262 |
| SatMAE | ViT-Base | 0.6325 | 0.6879 | 0.7355 | 0.6239 | 0.6980 | 0.7364 |
| Scale-MAE | ViT-Base | 0.6853 | **0.7333** | 0.7507 | 0.6910 | 0.7280 | 0.7512 |
| Cross-Scale MAE | ViT-Base | **0.7082** | 0.7225 | **0.7617** | **0.7163** | **0.7354** | **0.7603** |

the GSD positional encoding. Finally, we explore the effects of some hyper-parameters, including, for example, the number of epochs and the type of backbone used. All experiments in this section are conducted on the RESISC45 dataset [10] and use the KNN classification as a metric.

The weight update of the encoder and decoder networks is mainly controlled by the loss function in Eq. 6 that consists of three modules, the cross consistency loss in the encoder, $\mathcal{L}_{cc}$, the cross prediction loss, $\mathcal{L}_{cp}$, and the reconstruction loss, $\mathcal{L}_{re}$. Table 5 thoroughly compares the classification accuracy using different loss combinations, from which we make some interesting observations. This experiment is conducted on RESISC45 at two scaling ratios, $0.5, 1$, on the raw image with ViT-Base and pre-trained 300 epochs on fMoW-RGB dataset with input size of $128 \times 128$.

The first row is essentially SatMAE, with single scale input, and thus serves as the baseline for the subsequent comparisons. The second row adds the multi-scale input and solely uses reconstruction loss. The improved accuracy displays the effectiveness of multi-scale inputs, especially for lower-scale samples. From the second to the last row, all experiments use multi-scale inputs with different loss components. From the 3rd to the 4th row, where only one additional loss module is applied, we observe that both the consistency in the encoder representation and decoder representation are adequate, with the consistency in the decoder representation playing a more critical role. Finally, in the last row, we observe that the combination of consistency in the encoder and decoder has a performance increment of $3\% - 4\%$ on both scaling rates, again demonstrating the effectiveness of the proposed Cross-Scale MAE.

Second, we look into the effectiveness of negative samples in formulating the contrastive loss. Because Cross-Scale MAE leverages contrastive learning at two levels to handle multi-level representations, we must investigate the necessity of involving negative samples at each stage. Specifically, two kinds of losses are evaluated: InfoNCE relating to positive and negative samples and prediction distance relating to only positive samples. First, we assess it under the complete Cross-Scale MAE framework with two scaling ratios, and the results are reported in Table 6. From Table 6, we can find the negative samples in the encoder can improve the classification performance but decrease the performance when involved in the decoder. It may be because the representation from the encoder contains more

Table 5: Ablation study of the effect of each of the three losses in Eq. 6 (%)

| Multi-Scale | Cross-Consis | Cross-Pred | KNN 50% | KNN 100% |
|:---:|:---:|:---:|:---:|:---:|
| ○ | ○ | ○ | 52.1 | 58.9 |
| ✓ | ○ | ○ | 68.3 | 69.2 |
| ✓ | ✓ | ○ | 72.4 | 74.4 |
| ✓ | ○ | ✓ | 74.9 | 76.5 |
| ✓ | ✓ | ✓ | **78.7** | **79.3** |

Table 6: The effect of negative samples (NS) in Cross-Scale MAE (%)

| NS in Encoder | NS in Decoder | KNN 50% | KNN 100% |
|:---:|:---:|:---:|:---:|
| ○ | ○ | 75.9 | 77.9 |
| ✓ | ✓ | 76.8 | 77.7 |
| ○ | ✓ | 75.1 | 77.1 |
| ✓ | ○ | **78.7** | **79.3** |

"structural" information, but the decoder contains more "semantic" information [36]. Hence, the involvement of negative samples in the encoder can help boost the discriminative capacity between multi-scale inputs. On the other hand, with prediction loss between positive samples at the decoder, the consistency in "semantic" information can be better preserved. Having observed this interesting phenomenon, we are intrigued to study if the same pattern would be found in a pure contrastive learning (CL) framework, i.e., Cross-Scale MAE without the reconstruction loss. And we did, the results are shown in Table 7.

Third, we investigate the differences between scale augments and the GSD positional encoding proposed in Scale-MAE [36]. The results are presented in Fig. 3. Two sets of comparisons have been conducted – the left figure shows the performance of Cross-Scale MAE with or without the GSD positional encoding, and the right one shows the SatMAE performance with GSD position or multi-scale augments. The scaling ratios used in the comparison are $\{25\%, 50\%, 75\%, 100\%\}$. From Fig. 3(a), we find that Cross-Scale MAE adding GSD positional encoding decreases the performance. The reason may be that Cross-Scale MAE has already used multi-scale augmentation, rendering the GSD positional encoding redundant. On the other hand, from Fig. 3(b), we can find both GSD positional encoding and multi-scale augmentation can improve SatMAE's performance since SatMAE does not exploit cross-scale consistency in the images.

Finally, we show the performance of Cross-Scale MAE with different backbones at different training epochs and the effect of different masking ratios. The results are posted in supplementary materials.

### 4.3 Efficient Backbone

Large-scale models are slow to train and have high memory consumption. Thus, they are usually trained on large clusters of GPUs. We aim to make the model more accessible and optimize the implementation in various ways to minimize the training time and memory footprint for training and inference, such that the model can be feasibly trained end-to-end and used for inference on a single GPU while maintaining competitive results. In this work, we showed using the xFormers library to build an efficient foundation for our models, allowing us to train and conduct experiments more quickly.

Table 7: The effect of negative samples in the pure CL framework (%)

| NS in Encoder | NS in Decoder | KNN 50% | KNN 100% |
|:---:|:---:|:---:|:---:|
| ○ | ○ | 57.2 | 58.7 |
| ✓ | ✓ | 57.3 | 59.5 |
| ○ | ✓ | 57.1 | 58.2 |
| ✓ | ○ | **58.3** | **60.7** |

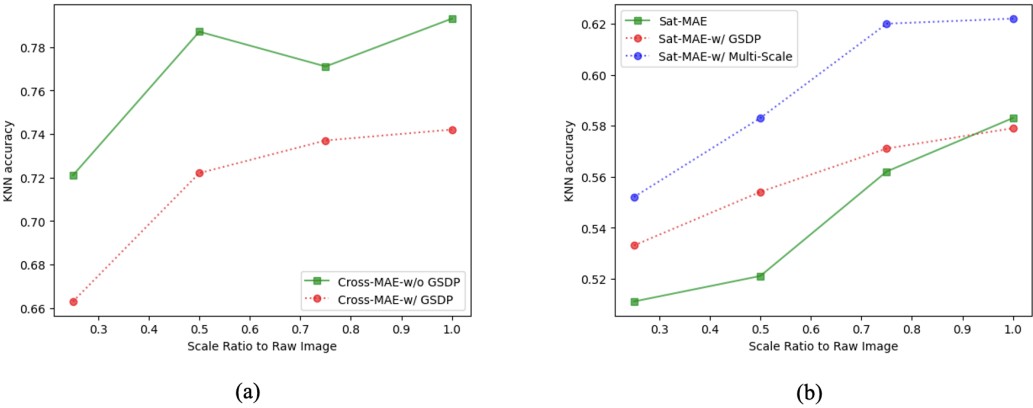

Figure 3: The comparison of multi-scale augmentation and GSD positional encoding.

Most of the optimizations offered by the xFormers library were made with Tensor-architecture GPUs in mind. We show that we can still gain noticeable improvements in training time even when using an older and less powerful GPU. Thus, for this comparison, we stress-test the xFormers and Timm versions of the baseline implementations by training them on individual Nvidia RTX A6000s. The details of the implementation and ablation studies on attention and loss types have been placed in the supplementary.

## 5    Conclusion & Future Work

This paper introduced Cross-Scale MAE, a novel multi-scale representation learning framework specifically designed for remote sensing image understanding. By incorporating scale augmentation and enforcing cross-scale consistency at both structural and semantic levels, our approach enhances the representational capabilities of the Masked Autoencoder in the context of multi-scale datasets. Remote sensing image analysis, which often relies on diverse multi-source data, greatly benefits from this framework. To achieve more robust and meaningful representations, we utilize both contrastive and generative losses, which encourage consistency and representation quality across different image scales. Furthermore, we employed the xFormers library to expedite network pre-training without sacrificing performance. Through extensive evaluation of various benchmarks, Cross-Scale MAE demonstrated superior performance compared to other state-of-the-art frameworks for satellite imagery.

Nonetheless, Cross-Scale MAE still has its limitations, which provide directions for our future work. (1) In scale augmentation, we currently focus on spatial augmentation (scaling) while maintaining consistent channel content across two augmentations. However, the complexity of remote sensing scenes extends beyond just scale differences. Variations in source images include both scale and channel disparities. We currently retain shared channels, like RGB, to address this challenge while discarding differing channels. This approach, although maintaining consistency, could lead to the loss of valuable information from dropped channels. For future work, we aim to investigate more into the multi-spectral perspective of representation learning. (2) Our multi-level contrastive learning employs diverse strategies across levels—leveraging both positive and negative samples at the encoder level and exclusively positive samples at the decoder level. This strategy currently yields optimal performance, although the underlying mechanism remains unexplored. In future research, we intend to investigate more into this strategy to gain deeper insights.

## Acknowledgements

This research is based upon work supported in part by the Office of the Director of National Intelligence (ODNI), Intelligence Advanced Research Projects Activity (IARPA), via 2021-20111000006. The views and conclusions contained herein are those of the authors and should not be interpreted

as necessarily representing the official policies, either expressed or implied, of ODNI, IARPA, or the U.S. Government. The U.S. Government is authorized to reproduce and distribute reprints for governmental purposes notwithstanding any copyright annotation therein.

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
