# Supplementary Materials for "Cross-Scale MAE: A Tale of Multiscale Exploitation in Remote Sensing"

Generally speaking, there are three ways to evaluate the effectiveness of a representation learning scheme: 1) use reconstruction error for a *direct* evaluation; 2) use ablation studies to investigate the impact of hyperparameter selection and different components of the proposed network structure; and 3) use downstream tasks like classification accuracy for an *indirect* evaluation. Since Cross-Scale MAE is essentially a self-supervised representation scheme, we follow similar standards. This supplement provides additional quantitative and qualitative (visual) results complementing results presented in the main paper to provide further validation of the claims made. Specifically,

- In Sec. A., we provide both visual and quantitative comparisons (in MSE and SSIM) of the reconstruction performance between baseline and Cross-Scale MAE.

- In Sec. B., we provide further ablation studies of Cross-Scale MAE on the effect of different masking strategies.

- An interesting question we asked ourselves is if the performance gain of the proposed Cross-Scale MAE also generalizes to natural images. In Sec. C., we investigate the performance of Cross-Scale MAE in natural images, using CoCo as training and test sets. We also study its generalization capacity across the different domains of natural imagery and remote sensing imagery.

- In Sec. D., we elaborate on the details of the deployment of xFormers as an efficient backbone (See Sec. 4.3 in the main paper) with further studies regarding the memory efficiency of different attention types.

## A. Direct Evaluation through Multiscale Reconstruction Performance

In this section, we analyze the multiscale reconstruction performance of the proposed Cross-Scale MAE. We compare it with SatMAE, a baseline model, and demonstrate the improvements achieved by Cross-Scale MAE. Both models were pre-trained on the fMoW-RGB dataset, and we assessed their capabilities in handling images at different scales.

To assess the reconstruction performance, we employ the Mean Squared Error (MSE) and the Structural Similarity Index (SSIM) [8, 5] metrics. These metrics quantify the structural difference between images, with a higher SSIM value and lower MSE value indicating better performance.

Fig. 1 demonstrates the improvements achieved across different scales. The first column represents the original input image before masking, while the second column displays the input with a 75% mask applied. The third column exhibits the reconstruction results of the baseline model, while the last column showcases the improved reconstruction achieved by our model, Cross-Scale MAE. Additionally, the corresponding SSIM metric is presented alongside each reconstruction. The red and green boxes over the raw image in the first row showcase the crop locations of the images in the second and third rows, respectively.

We observe a significant reduction in artifacts within the masked portions of the reconstructed images. The artifacts in these regions indicate the presence of uninformative or distorted latent representations. This observation implies that such regions would be ineffective for representation learning and might even negatively impact downstream models attempting to learn from these representations. Notably, at the full scale (first row), our model demonstrates a 38% improvement in the SSIM metric. At a 40% scale (second row), we observe an 18% improvement. Finally, at a 25% scale (last row), our model showcases a 35% improvement.

To further illustrate the effectiveness of Cross-Scale MAE, we provide additional visualizations in Fig. 2. These samples showcase individual images cropped to random scales, highlighting the superior reconstruction achieved by our model compared to the baseline.

To comprehensively evaluate the performance in a multiscale scenario, we present the average metrics over the fMoW-RGB testing set in Fig. 3. Each input image is evaluated 25 times with different random crop scales and masks. This procedure ensures a robust assessment of our model's performance in a multiscale context.

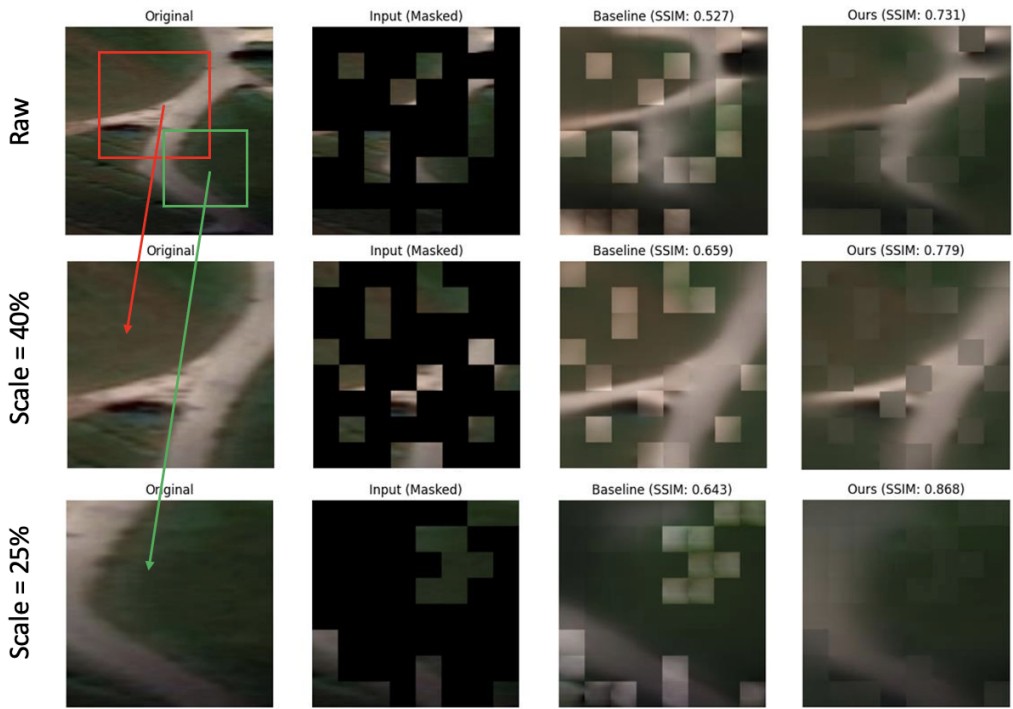

Figure 1: Comparison of Cross-Scale MAE and SatMAE reconstructions at different scales

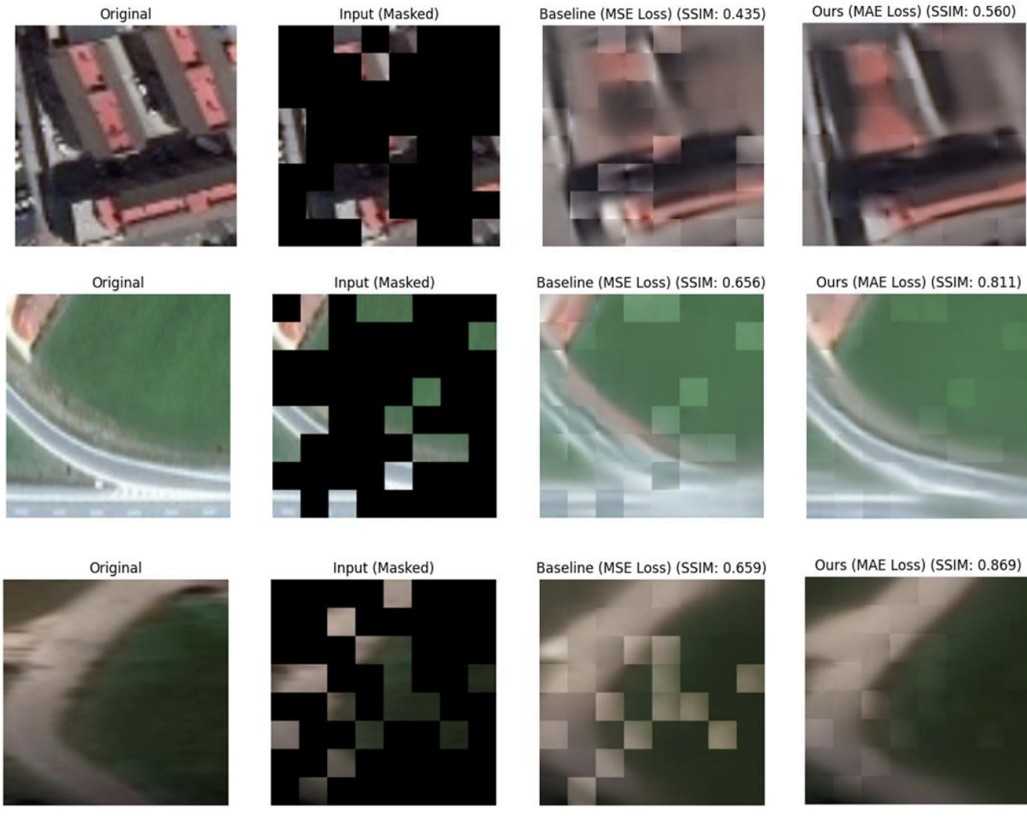

Figure 2: Reconstruction samples with random scales (fMoW test set)

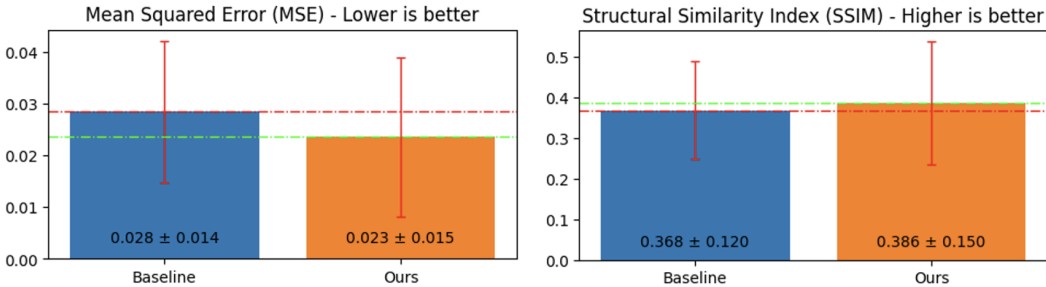

Figure 3: Average metrics comparison of Cross-Scale MAE and SatMAE (fMoW test set)

Our extensive evaluation demonstrates that Cross-Scale MAE excels in reconstructing multiscale images, surpassing the performance of the baseline model, SatMAE. The observed improvements and the mitigation of artifacts in masked portions indicate the superiority of Cross-Scale MAE in capturing meaningful representations and enhancing remote sensing image understanding in diverse scale conditions.

## B. More Ablation Study

In this section, we investigate the impact of mask consistency and masking ratios on the model's representation learning capacity.

To assess the quality of the learned representations, we employ a non-parametric k-Nearest Neighbor (kNN) classification approach with zero-shot learning. This evaluation method measures the ability of the pre-trained model to produce semantically coherent representations, a desired characteristic for practical zero-shot classification tasks. Similar evaluation strategies have been employed in other notable works [9, 3, 2].

In the following subsections, we conduct detailed analyses to examine the effects of mask consistency and masking ratios on the performance of Cross-Scale MAE. We employ the RESISC dataset and utilize the ViT-Base as the backbone architecture for the evaluations, same as used in the main paper.

### B.1. Effect of Mask Consistency

In Cross-Scale MAE, we generate two scale augments from the raw image and have the option to apply either a consistent mask, where the patch location of the mask remains fixed across both scale images, or different masks with varying patch locations for each scale. In this section, we compare the performance of these two cases using kNN with $k=20$ on the representations with different scale ratios.

Table 1: Effect of mask consistency in Cross-Scale MAE on RESISC

| Masking Strategy | kNN 25% | kNN 50% | kNN 100% |
|------------------|---------|---------|----------|
| Consistent       | 0.762   | 0.812   | 0.824    |
| Different         | **0.787** | **0.831** | **0.853** |

Table 1 presents the evaluation results, showcasing the effect of mask consistency on Cross-Scale MAE performance on the RESISC dataset. The table highlights the kNN accuracy for different scale ratios. Notably, we observe that inconsistent masks yield nearly a 2% improvement in performance. This improvement may be attributed to the introduction of additional variance during training, resulting in a more robust and invariant representation being learned by the model.

## B.2. Effect of Masking Ratio and Training Time

In addition to exploring the impact of mask consistency, we also investigate the effect of different masking ratios on the performance of Cross-Scale MAE. Table 2 reports the evaluation results using three different mask ratios: 60%, 75%, and 90%. The kNN accuracy for each mask ratio at various scale ratios is measured and compared.

Table 2: Effect of Mask Ratio in Cross-scale MAE on RESISC

| Masking Ratio | KNN 25% | KNN 50% | KNN 100% |
|---|---|---|---|
| 60% | 0.7803 | **0.8322** | 0.8407 |
| 75% | **0.787** | 0.831 | **0.853** |
| 90% | 0.7524 | 0.7971 | 0.7977 |

We observe interesting patterns from the results presented in Table 2. A relatively low mask ratio of 60% still yields excellent performance for remote sensing images, demonstrating competent representation learning capabilities. However, employing a high mask ratio of 90% leads to decreased performance. This reduction may be attributed to the significant information loss caused by a high degree of masking, which affects the model's ability to capture essential details and features.

These findings highlight the importance of carefully selecting an optimal mask ratio to balance preserving relevant information and encouraging robust representation learning.

Finally, we show the performance of Cross-Scale MAE with different backbones at different training epochs in Table 3.

# C. Evaluation on Natural Images (CoCo Dataset)

To assess the generalization capabilities of Cross-Scale MAE, we evaluate by pre-training on the CoCo2017 dataset, focusing on natural images. This evaluation allows us to validate the effectiveness of our model beyond remote sensing images and examine its performance in a different domain. We pre-train the Cross-Scale MAE on the CoCo2017 dataset using the following parameter settings: ViT-Base as the backbone architecture, a base learning rate of $5 \times 10^{-3}$, a weight decay of $5 \times 10^{-3}$, and an input size of $128 \times 128$. The model is trained for 400 epochs.

## C.1. Pre-Training Performance on CoCo Images

We visualize the reconstruction results on the CoCo dataset in Fig.4 and present the corresponding evaluation metrics in Fig.5. We compare the performance of Cross-Scale MAE with the baseline model, MAE [4].

From Fig.5, we observe that Cross-Scale MAE outperforms the baseline model in terms of both MSE and SSIM. Additionally, in Fig.5, we notice that the baseline model exhibits more artifacts in the reconstruction results at the locations of the masked patches. In contrast, Cross-Scale MAE demonstrates a closer representation of the actual distribution of pixel values that should be present in those locations.

## C.2. Zero-Shot Performance on fMoW-RGB Images

Furthermore, we evaluate the zero-shot reconstruction performance of Cross-Scale MAE on the fMoW-RGB dataset using the model pre-trained on CoCo. In this evaluation, we freeze the model trained on CoCo and reconstruct images from the fMoW-RGB dataset. It is important to note that the model has not seen any images from the fMoW-RGB dataset during its training. The zero-shot

Table 3: Performance with different backbone and training epoch of Cross-Scale MAE (%)

| Epochs | 50 | 100 | 150 | 200 | 250 | 300 |
|---|---|---|---|---|---|---|
| ViT-Base | 63.72 | 74.13 | 75.42 | 77.73 | 78.55 | 79.25 |
| ViT-Large | 60.01 | 73.84 | 79.42 | 83.09 | 83.39 | 85.34 |

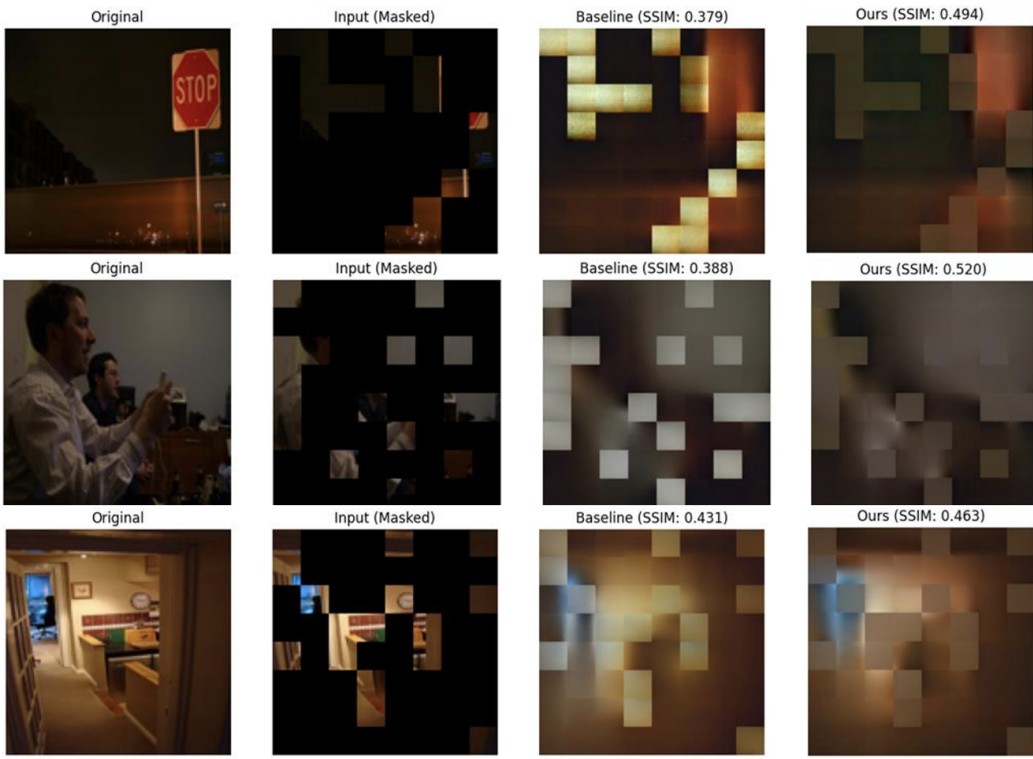

Figure 4: Reconstruction performance on the CoCo Dataset

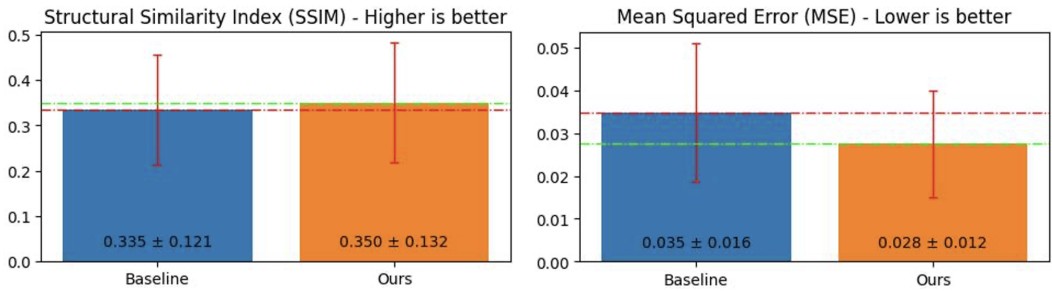

Figure 5: Multiscale reconstruction performance on the CoCo dataset

reconstruction results are displayed in Fig. 6. These results indicate that the learned representations by Cross-Scale MAE generalize well, as the zero-shot reconstruction still produces meaningful outputs on the fMoW-RGB dataset.

The evaluation of natural images demonstrates the versatility of Cross-Scale MAE, showcasing its ability to capture meaningful representations and generalize effectively across different domains. These findings highlight the potential of our model to enhance image understanding and reconstruction tasks in various applications beyond remote sensing imagery.

## D. Timm vs. xFormers MAE Backbones

This section presents the findings of an ablation study conducted as the initial step of our research. The study aimed to establish an efficient and flexible backbone for the final implementation by optimizing the original Masked Auto-Encoder for improved training time and a smaller memory footprint. The ultimate goal was to enable feasible end-to-end training and inference on a single GPU.

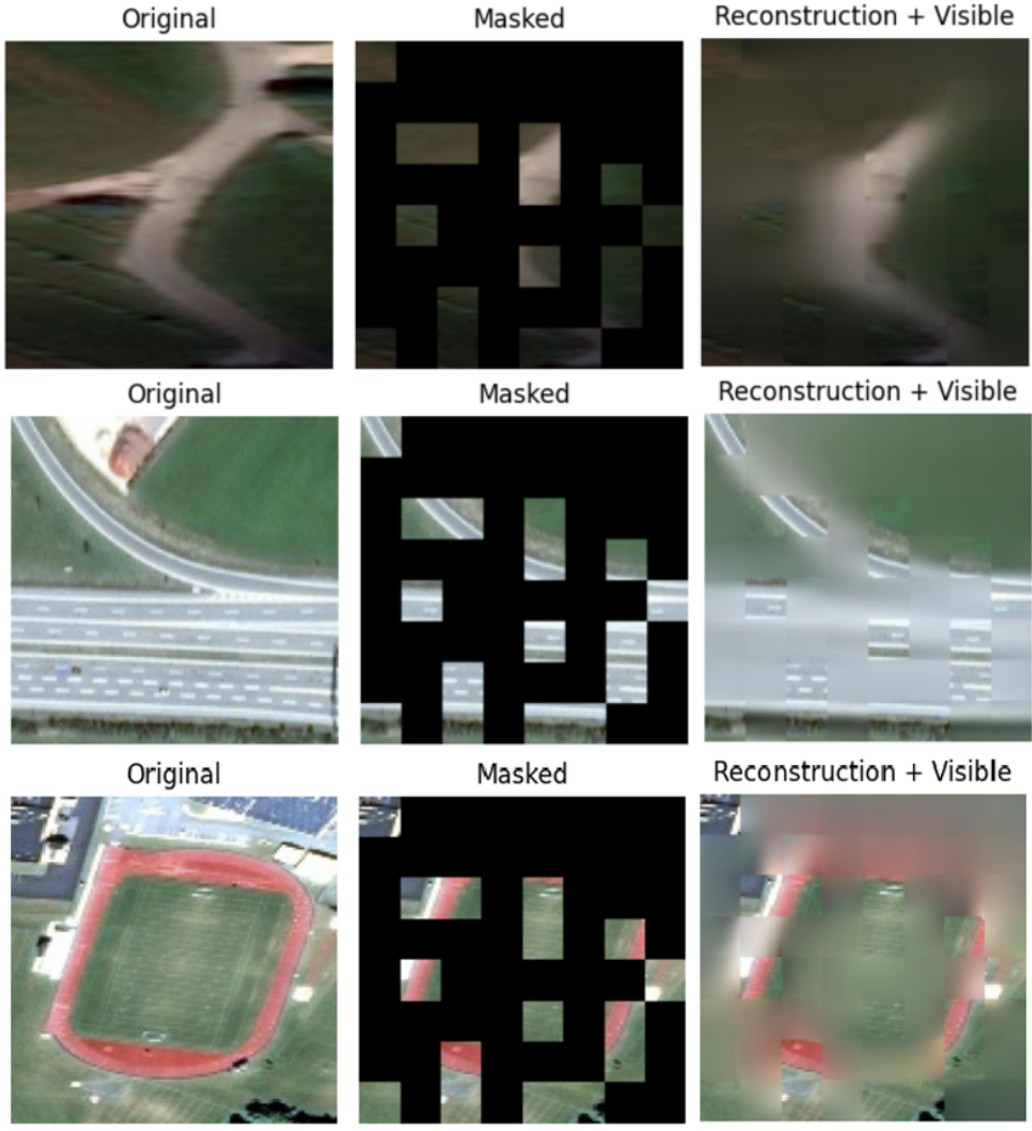

Figure 6: Zero-shot reconstruction in fMoW with Cross-Scale MAE pre-trained on CoCo

Initially, the MAE implementation relied on the Timm library for its various components. However, during our research, we came across the xFormers[7] library, which offers customizable building blocks and cutting-edge components yet to be available in mainstream libraries like PyTorch. xFormers claims to be built with efficiency in mind, delivering fast and memory-efficient performance[7].

## D.1. Baseline Efficiency Benchmark

We re-implemented the original MAE using this library to evaluate the potential benefits of using xFormers. This allowed us to compare the performance of the Timm implementation against the xFormers implementation based on PyTorch 1.13.1. Additionally, we considered the recently released PyTorch 2.0.0, which promised improved efficiency and optimizations for transformer models. It is important to note that xFormers did not yet support the newly released PyTorch version during our experiments.

We conducted several benchmark experiments to evaluate the performance of the original Timm implementation using PyTorch 1.13.1 and PyTorch 2.0.0 against our xFormers implementation on PyTorch 1.13.1. These evaluations were performed on an NVIDIA RTX A6000 GPU.

Table 4: Time/Step - 224 × 224, patch16, batch256

| Model | Timm (1.13.1) | xFormers (1.13.1) | Timm (2.0.0) |
|-------|---------------|-------------------|--------------|
| ViT-Base | 0.4849 | **0.4144** | 0.4685 |
| ViT-Large | 0.7939 | **0.7143** | 0.7584 |

Table 5: Memory Usage - 224× 224, patch16, batch256

| Model | Timm (1.13.1) | xFormers (1.13.1) | Timm (2.0.0) |
|-------|---------------|-------------------|--------------|
| ViT-Base | 22639 | **19020** | 22223 |
| ViT-Large | 34225 | **30739** | 32974 |

The results revealed the superior performance of xFormers when working with an input resolution of 224x224. Comparing memory usage and time per step, xFormers outperformed the Timm implementation on PyTorch 1.13.1. Specifically, the Vision Transformer (ViT) Base achieved a 17% increase in speed, while ViT Large demonstrated an 11% improvement with xFormers. Regarding memory efficiency, xFormers showcased a 19% enhancement for ViT Base and an 11% improvement for ViT Large. It is worth noting that PyTorch 2.0.0 also provided some speed and memory improvements at this input resolution, often falling between the performance of Timm and xFormers on PyTorch 1.13.1.

Table 6: Time/Step - 128 ×128, patch16, batch512

| Model | Timm (1.13.1) | xFormers (1.13.1) | Timm (2.0.0) |
|-------|---------------|-------------------|--------------|
| ViT-Base | 0.2948 | 0.2820 | **0.2796** |
| ViT-Large | 0.5245 | 0.5047 | **0.4986** |

Table 7: Memory Usage - 128 ×128, patch16, batch512

| Model | Timm (1.13.1) | xFormers (1.13.1) | Timm (2.0.0) |
|-------|---------------|-------------------|--------------|
| ViT-Base | 12213 | 12003 | **11805** |
| ViT-Large | 20891 | 21060 | **19601** |

When utilizing an input size of $128 \times 128$, the performance differences were subtle. The original Timm implementation on PyTorch 2.0.0 exhibited a slight advantage over the xFormers implementation on PyTorch 1.13.1. The time per step showed a mere 1% improvement with the Timm implementation on PyTorch 2.0.0. In terms of memory usage, there was an approximate 7% improvement with the Timm implementation on PyTorch 2.0.0 compared to the xFormers implementation on PyTorch 1.13.1. Although present, these differences were less substantial than those when using a higher input resolution.

The results of the ablation study underscored the significance of using xFormers as the backbone for our final implementation. Not only did xFormers provide enhanced flexibility through its customizable building blocks and cutting-edge components, but it also demonstrated superior speed and memory efficiency performance.

## D.2. Effect of Attention Type

Our experiments primarily used the Scaled Dot Product (SDP) attention mechanism, a common choice for transformer architectures. However, attention mechanisms can significantly influence a model's efficiency and reconstruction accuracy. The initial implementation, using the Timm library, only supported SDP attention. In contrast, the xFormers library—our final choice for implementation—provides an expanded selection of attention mechanisms, allowing for a more comprehensive examination of how different attention types affect model performance and efficiency.

The Fourier Mix attention [6], which integrates the generalized Fourier integral theorem into the dot-product attention step of the standard transformer, showed significant improvements in both speed and memory consumption. Compared to SDP attention, Fourier Mix attention was 34% faster and consumed approximately 44% less memory. Incorporating Fourier Mix attention addresses the

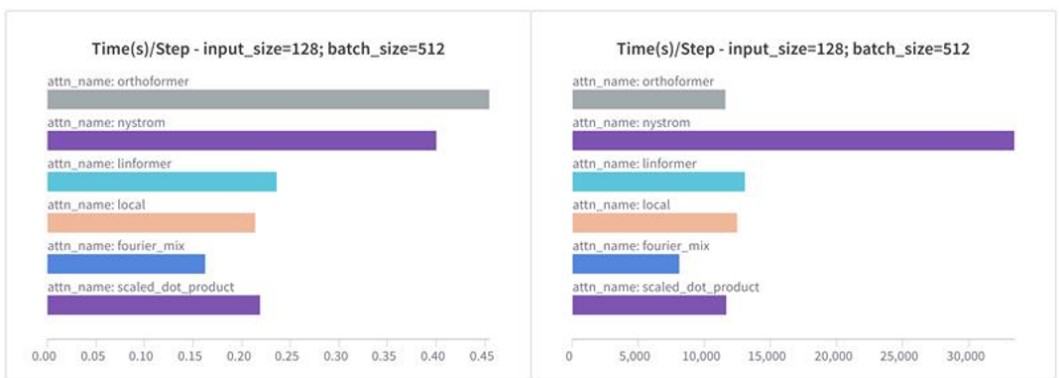

Figure 7: Memory efficiency of different attention types

traditional SDP attention's limitations, capturing complex interactions among the features of the queries and keys more effectively and reducing redundancy between attention heads [6].

On the other hand, the Local attention [1] mechanism provided only a minor speed improvement but also slightly increased memory consumption for our current architecture. Local attention offers a novel approach to managing long sequences by dividing attention into global and local components to facilitate the efficient processing of long input sequences. Due to this, we suspect this mechanism's benefits would be a lot more noticeable in even larger architectures with much larger embedding dimensions.

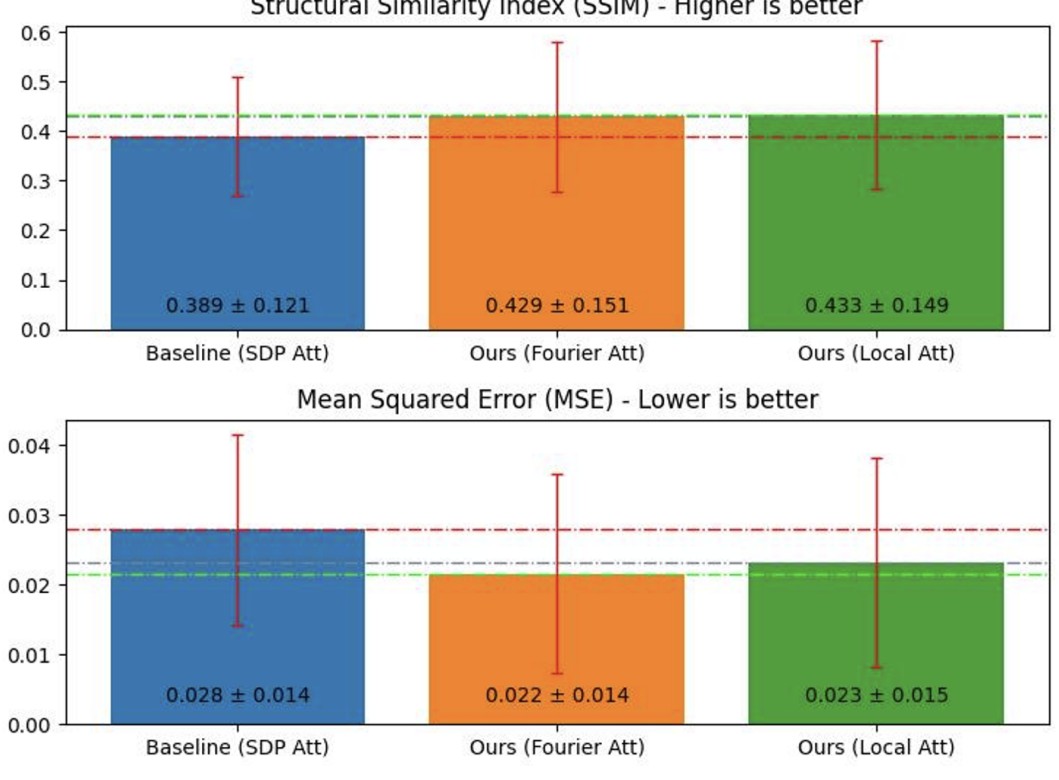

Figure 8: Performance comparison of attention types in Cross-Scale MAE

Following these observations, we further tested the Fourier Mix and Local attention mechanisms, assessing their reconstruction performance against the SDP attention baseline. As shown in Figure 8, both Fourier Mix and Local attention mechanisms demonstrated a 10% improvement in the Structural Similarity Index (SSIM) metric, a method for comparing image similarities essential for multiscale performance. These attention mechanisms also significantly improved the Mean Squared Error (MSE) metric, which quantifies the average squared differences between estimated and actual values.

Our findings underscore the potential of exploring alternative attention mechanisms to enhance efficiency and performance. The xFormers library, with its diverse attention options, provides an opportunity to tailor attention mechanism selection to specific applications, leading to substantial performance gains.