# OpenReview forum: "Cross-Scale MAE: A Tale of Multiscale Exploitation in Remote Sensing"
_NeurIPS.cc/2023/Conference — NeurIPS 2023 poster_

### Official Review · Reviewer_14Vd · 2023-06-21

**Soundness:** 3 good
**Presentation:** 3 good
**Contribution:** 3 good
**Rating:** 7
**Confidence:** 4

**Summary:**

The authors proposed a self-supervised method based on ViT masked auto encoders (MAE), namely Cross-Scale MAE, to improve the representations learnt in remote sensing based models. It includes a scale augmentation of each input image to learn features at different scales in the encoder while overcoming the need of multiscale aligned data. A constraint is also imposed by exploiting information consistency across multiscale images and improving structural and semantic level representation.
The proposed Cross-Scale MAE methods includes both discriminative and generative approaches.
Experiments have shown better performances on downstream tasks compared to remote sensing self-supervised MAE methods, namely SatMAE and Scale-MAE. The model has been optimized using xFormers to improve its computational efficiency without performance degradation.

**Strengths:**

1/ The Cross-Scale MAE method has been clearly explained. It combines both discriminative and generative approaches to learn scale-invariant representations. The cross-scale consistency loss aims to maximize information at different scales while minimizing the sharing information between images at different locations. The cross-scale prediction and reconstruction losses improve the consistency and effectiveness of the decoder to retrieve the semantic information. This original combination is agnostic to the backbone used. It may tackle the actual multiscale problems related to various resolutions in satellite data.

2/  The presented results have shown that Cross-Scale MAE outperforms competing methods on several datasets according the KNN classification accuracy with representation as a metric highlighting the relevance of the learnt presentations.
The ablation studies have been rigorously conducted showing the impact of each proposed loss term and the global combination, the effect of the contrastive loss used to constrain the encoder and the influence of the scale augmentation methods.

3/ Experiments on the computational time and memory usage conducted in the Appendices are appreciated.

4/ This work could have a significant impact since large volume of remote sensing data are publicly available without labels. There is an urgent need of efficient and powerful pretrained models for many remote sensing applications, including climate change and global warming understanding and monitoring.


**Weaknesses:**

1/ Additional experiments presented in the appendices have led to similar conclusions than presented in the main document. However, one may notice that the proposed method outperforms SatMAE in average but not consistently considering the presented standard deviations in Appendices Figure 3, Figure 5 and Figure 8. Standard deviations in the main document, including competing methods, would have been appreciated to highlight the consistency of the results.

2/ L231 "SatMAE is a current state-of-the-art Masked Auto-Encoder (MAE) for remote sensing imagery" The GFM model [1] could also be considered as a strong MAE competing method outperforming SatMAE. This method has not been compared to Scale-MAE which could be interesting for the community and could potentially highlight the relevance of the proposed method.


[1] M. Mendieta et al., GFM: Building Geospatial Foundation Models via Continual Pretraining. In Arxiv 2023.

**Questions:**

Questions

1/ Figure 3: why performance curves of Scale-MAE are not included?

2/ The experiments have been conducted considering several fixed scale ratios. Have the performances been quantified using a random scale ratio at each batch during training?


Comments

1/ Figure 2: AT blocks are not defined, it is not clear in (b) what $f_{e,n}$ is.

2/ Typo: lack of consistency between "multiscale" and "multi-scale"

3/ Typo: lack of consistency between "SatMAE" and "Sat-MAE"

4/ Typo: $\tau$ in Eq. 3 is not defined.


**Limitations:**

The authors have neither detailed the limits, nor the potential negative societal impacts (military or surveillance applications) of the proposed method. The publication of the code and the pre-trained model was not mentioned eithe and will condition the final rating of the submission for reproducibility reasons and the accessibility to the public.

---

> ### Author Rebuttal · Authors · 2023-08-10
>
> Thanks for your meaningful comments. Regarding to your question and comments, our answers are listed as following.
>
> W1. On moving figures from Supplement to the main text. Thanks for the suggestion. We will move one of these figures to the main text and highlight the following: The graphs in the supplementary material are generated by conducting 25 runs for each sample in the testing dataset. In each of these 25 runs for a specific sample, the sample is cropped to a different random scale before being input into the models. For an individual run, the same image scale and masking configuration are used as inputs to the models being compared to ensure a fair comparison.
>
> W2.  On comparison to GFM. The GFM model excels in representation learning through continual learning, aiming to enhance large language model applicability to satellite images via knowledge distillation. Although GFM's source code is not yet available, we recognize its reported segmentation mIoU of 0.753 on the Vaihingen dataset. In a rough comparison, Cross-Scale MAE achieved an accuracy of 0.7603 on the same task. We will include the GFM paper in our related work for comprehensive context.
>
> Q1. On missing Scale-MAE in Fig. 3. Thanks for pointing this out. Scale-MAE was not open source until recently. We are thus able to add its performance to Fig. 3. Please check Fig. 2 of the rebuttal.
>
> Q2.  On using random scale ratio. Yes, indeed! The model randomly selects a scale ratio from 0.2-0.8 at each batch during the pre-training.
>
>
> To the comments:
> 1. We have improve the model structure figure(figure.2 of the paper) and posted in the rebuttal file as figure.3, please check.
>
> 2-4. Thanks for pointing out the typos, we will correct them in the revision.

---

> > ### Comment · Reviewer_14Vd · 2023-08-12
> >
> > I would like to thanks the authors for their valuable rebuttal.
> >
> > The additional results and materials will improve the submission significantly, including new results comparing to other SSL methods and extensive comparisons with the competing methods.
> > The updated architecture figure is also appreciated.
> >
> > Considering this additional work and the answers provided to the other reviewers, I would be inclined to increase my rating towards the acceptance.
> >
> > However, it will be conditioned to the concern mentioned in "limitations" which have not been mentioned by the authors.  "The publication of the code and the pre-trained model was not mentioned either and will condition the final rating of the submission for reproducibility reasons and the accessibility to the public."
> > Could you please provide a feedback on this topic?

---

> > > ### Author Response · Authors · 2023-08-13
> > > **Reply to the Limitations**
> > >
> > > Thank you for swiftly providing feedback on our rebuttal. To your concerns, we answer as following:
> > >
> > > Regarding the limitation and next step:
> > >
> > > (1). In our augmentation, we currently focus on spatial augmentation (scaling) while maintaining consistent channel content across two augmentations. However, the complexity of remote sensing scenes extends beyond just scale differences. Variations in source images include both scale and channel disparities. To address this challenge, we currently retain shared channels, like RGB, while discarding differing channels. This approach, although maintaining consistency, could lead to loss of valuable information from dropped channels. In forthcoming endeavors, we aim to devise solutions for this issue.
> > >
> > > (2). Our multi-level contrastive learning employs diverse strategies across levels—leveraging both positive and negative samples in the encoder level, and exclusively positive samples in the decoder level. This strategy currently yields optimal performance, although the underlying mechanisms remain unexplored. In future research, we intend to delve into the intricacies of this strategy to gain deeper insights.
> > >
> > > Regarding the publication of the code and the pre-trained model:
> > >
> > > Thank you for inquiring. We are strong advocates of open source principles, and we are committed to sharing our code and pre-trained models on GitHub following the submission.

---

### Official Review · Reviewer_LSpN · 2023-07-01

**Soundness:** 3 good
**Presentation:** 3 good
**Contribution:** 3 good
**Rating:** 6
**Confidence:** 3

**Summary:**

This paper proposed a flexible self-supervised learning (SSL) framework that yields robust representations named Cross-Scale MAE by enforcing cross-scale information consistency at structural and semantic levels without needing aligned multiscale remote sensing imagery. And this paper deploys xFormers to realize the Cross-Scale MAE.

**Strengths:**

1. This paper designed Cross-Scale MAE to enhance the consistency of the representations obtained from different scales for the multi-scale objects in remote sensing.
2. This paper deploys xFormers to realize the Cross-Scale MAE, which provides a viable reference for other models to be trained in a single GPU.


**Weaknesses:**

Lack of clarity in expressing the specific problem to be solved by the designed model

**Questions:**

1. What exactly is the semantic understanding of remote sensing image understanding, and are there any specific applications？ such as remote sensing image classification, and remote sensing image change detection,...

2. In Section 4, are there some visualization results and the related analysis to show the advantages of the Cross-Scale MAE?

3. In Section 4, lines 235-238, “In Section 5.1” and “Section 5.2” have errors.

---

> ### Author Rebuttal · Authors · 2023-08-10
>
> Thanks for your meaningful comments. Regarding to your question and comments, our answers are listed as following.
>
> To the questions:
>
> Q1. On semantic understanding: We would like to explain our viewpoint by addressing them at two levels of representations used in the loss. In the paper, we used two contrastive losses at different levels of representation; one is the output of the encoder (feature 1), and the other is the output after layernorm of the last self-attention block in the decoder (feature 2), which is used to map to the reconstructed image. Comparing the two representations, we could say that feature 1 is a relatively 'lower' feature than feature 2 and thus contains more structural information. Feature 2 is relatively 'closer' to the reconstructed image, which includes more semantic information than feature 1. We will clarify this by emphasizing that this is a “relative” definition.
>
> Due to the usage of cross-scale contrastive loss at the encoder and a cross-scale predictive loss at the decoder, the extracted representation would possess both discriminative and representative characteristics. Hence, the learned representation can serve well for downstream tasks like classification and segmentation. In the Supplement, we posted more results, including the downstream classification task (Supplement C) and zero-shot transfer learning performance on CoCo (Supplement D). We have since added comparisons with more models and on more downstream tasks. In the rebuttal file, please refer to Table 1 (for comparing segmentation performance with SOTA self-supervised segmentation approaches).
>
>
> Q2. On visualization results: We have included visualization results and related analyses in the Supplement. Supplement A (Figures 1 & 2) provided visualizations of the advantages of our method pre-trained on the fMoW dataset. We also showcased results from a model pre-trained on the CoCo dataset in Figure 4 of Supplement D.1. Finally In Supplement D.2., Figure 6, we provided a visualization of the zero-shot performance of our method where a model pre-trained on the CoCo dataset is tested against samples from the fMoW test set, without any additional fine-tuning on CoCo.
>
> Q3. Thanks for pointing out the typos made in Sec. 4. We will make sure to correct it in the revision.

---

> > ### Comment · Reviewer_LSpN · 2023-08-15
> > **Final Rate**
> >
> > Thank you for your response,  I have no further questions and will maintain my rating.

---

### Official Review · Reviewer_TztZ · 2023-07-04

**Soundness:** 3 good
**Presentation:** 2 fair
**Contribution:** 2 fair
**Rating:** 3
**Confidence:** 4

**Summary:**

This paper proposes a Cross-Scale MAE to tackle the multiscale problem in remote sensing images. The triplet loss is designed including cross-scale consistency loss at the encoder, cross-scale prediction loss at the decode, and reconstruction loss. Comparative experiments show competitive performances.

**Strengths:**

1. This paper introduces the cross-scale consistency for image representation.
2. The scales are aligned in both the encoder and decoder.
3. Compared experiments show its superiority.


**Weaknesses:**

1. The authors claim the semantics have been aligned in this method. However, the object semantics described in Fig.1 are vague. Because, there are no semantic inputs or guidance (object labels, etc.). Therefore, it is inaccurate to claim that the feature output by the decoder is the semantics. Normally, decode is used for image reconstruction tasks (Fig.2), and the obtained features should be the low-level structure information.
2. Multiscale is an old research topic. In this paper, methods such as MAE and contrastive learning are applied to align the multi-scale enhanced inputs, most of these techniques exist in contrastive learning. I am not optimistic about the novelty of the paper.
3. In the experiment, it is only compared with the MAE method in remote sensing. Other contrastive learning methods and the MAE method in machine learning have not been discussed yet.
4. The experimental accuracies in table 2 should align with their original papers (Sat-MAE, Scale-MAE). Only K-NN testing seems not enough.


**Questions:**

Fig.1 can not convey effective information. It just shows different sizes of aircraft, common sense, and very little useful information. I think this picture can be integrated into your research motivation or method innovation, not just three remote-sensing images.

---

> ### Author Rebuttal · Authors · 2023-08-10
>
> Thanks for your meaningful comments. Regarding to your question and comments, our answers are listed as following.
>
> W1. On “semantic” vs. “structural”: We agree with the reviewer about the meaning of low-level features. We would like to explain our viewpoint by addressing them at two levels of representations used in the loss. In the paper, we used two contrastive losses at different levels of representation; one is the output of the encoder (feature 1), and the other is the output after layernorm of the last self-attention block in the decoder (feature 2), which is used to map to the reconstructed image. Comparing the two representations, we could say that feature 1 is a relatively 'lower' feature than feature 2 and thus contains more structural information. Feature 2 is relatively 'closer' to the reconstructed image, which includes more semantic information than feature 1. We will clarify this by emphasizing that this is a “relative” definition.
>
> W2. On the novelty of the paper: To clarify, we are not 'aligning' multi-scale inputs, but extracting shared representations. Scale augmentation guarantees alignment.
>
> While our pipeline may seem straightforward at first glance, it solves three challenging problems. First, given that masking in MAE can be interpreted as a type of augmentation, are there other types of augmentations that can enhance the robustness of representation learning? This paper figures out an intuitive scale augmentation approach that enables scale-invariant representation. Our ablation study (Table 3 of the main text) showed the significant performance gain between using scale augmentation (2nd row) and without it (1st row).
>
> Second, given the multiple-scale inputs, most existing pipelines would extract features independently from these inputs without exploiting the cross-scale correlation. In the proposed pipeline, we designed three losses that exploit this cross-scale correlation at two levels, the cross-scale consistency loss at the encoder embed and the cross-scale predictive loss at the decoder embed. In fact, how to use the contrastive loss to calculate the cross-scale consistency loss is a contribution of its own. The ablation study in Table 3 of the main text also showed the significant contribution of each of the cross-scale loss function (Table 3, rows 3-5).
>
> Third, the incorporation of xFormers into our pipeline enables the pre-training process to be performed on single GPU, providing a practical pipeline for ViT-based model training. For additional xFormer insights, please refer to Supplement E.
>
> Essentially, compared to SatMAE which is the first paper that applies MAE to extract representations from satellite images with single and fixed scale, Scale-MAE and the proposed Cross-Scale MAE are also based on MAE, but focus on the multi-scale problem. Specifically, Scale-MAE develops the Ground Sample Distance (GSD) position encoding and applies multi de-convolution after decoder to reconstruct images of different scales. Nonetheless, Scale-MAE integrates the scale information into the network via hard coding of known GSD. And, the de-convolution can only result in a specific scale ratio. The proposed Cross-Scale MAE designs the network to learn the information across different scales. With scale augmentation and multi-level contrastive loss between the scale pair and masked patches reconstruction, Cross-Scale MAE can learn informative and consistent representation across different scales without the need of known GSD.
>
>
> W3. On comparing with other contrastive learning and MAE methods: We have added more comparisons in the rebuttal file, including more contrastive learning frameworks and MAE-based framework. Please check Tables 1 and 2 of the rebuttal file.
>
> W4. On KNN testing being insufficient. We completely agree that having kNN evaluation is insufficient. Due to the usage of cross-scale contrastive loss at the encoder and a cross-scale predictive loss at the decoder, the extracted representation would possess both discriminative and representative characteristics. Hence, downstream tasks should include both classification and segmentation performance. We have since conducted additional experiments on segmentation that further demonstrate consistent improvements over SOTA self-supervised segmentations. The results are shown in Table 1 of the rebuttal.
>
> In addition, the Supplement includes more results, such as the downstream classification task (Supplement C) and zero-shot transfer learning performance on CoCo (Supplement D).
>
> Q1.On Figs. 1 and 2: Thanks for the suggestion. We have redrawn Fig. 2 and combined information from Fig. 1. Please see Fig. 3 in the rebuttal file for a revised illustration of the system architecture.

---

> ### Comment · Reviewer_TztZ · 2023-08-15
> **Final Rating**
>
> The final rating is Reject.
> As a pretrained model, the transferability in diverse downstream remote sensing tasks is significant; however, this manuscript only presented the image classification performance. The claim of superior to SatMAE and Scale-MAE is not suitable. Please check their settings; they adopted their pretrained MAE on various remote sensing tasks, not only image classification.
>
> I agree with Reviewer UGSM. This manuscript presented a good practice of combining conventional algorithms and validated their approach to image classification. Thank the authors for their effort. However, it brings little knowledge improvement to me. I also agree the Reviewer 14Vd, we have our own criteria for novelty. I don't think this combination can meet the standard of NeurlPS's audience.
>
> Besides, for downstream results in Supplementary, I have some suggestions for authors to improve this manuscript.
> First, reconstruction performance may be the result we do not really concern about. For SSL, transferred performances on the downstream and non-pretext tasks should be our focus. Better reconstruction performance makes no sense here. But I appreciate the reconstruction results the authors showed.
>
> Given the limited novelty and insufficient downstream transfer experiments, I have to reject this manuscript in this round, though I think this manuscript is promising if the authors can provide desired experiments and more thoughtful theoretical analysis.
>
> Thank the authors for their manuscript and detailed rebuttals.

---

### Official Review · Reviewer_UGSM · 2023-07-05

**Soundness:** 2 fair
**Presentation:** 2 fair
**Contribution:** 2 fair
**Rating:** 3
**Confidence:** 5

**Summary:**

This paper presents Cross-Scale MAE, a self-supervised model built upon the Masked Auto-Encoder (MAE), which tackles the challenges in remote sensing image understanding (such as extensive coverage, hardware limitations, and misaligned multiscale images) by learning relationships between data at different scales during pretraining. By introducing scale augmentation and enforcing information consistency across multiscale images at both structural and semantic levels, the proposed model demonstrates improved performance in downstream tasks compared to standard MAE and other existing remote sensing MAE methods.

**Strengths:**

Cross-Scale MAE combines discriminative and generative learning approaches, benefiting from self-supervised and multiscale representation learning advances. The novel approach of scale augmentations and multi-level cross-scale consistency constraints ensures consistent and meaningful representations, enabling enhanced remote sensing image understanding.

**Weaknesses:**

Some points are confusing, which prevents the readers from understanding the main ideas. Also, the compared methods are not enough to illustrate the advantages of the proposed cross-scale MAE.

**Questions:**

1.	Introduction should be simplified. The main contributions and motivation should be highlighted as clearly and simply as possible.
2.	From the current version, I just learn that the authors combine some existing techniques to handle their tasks, limiting the novelty of the proposed model.
3.	The caption of Fig. 2 is incomplete and meaningless. Also, the information conveyed by Fig. 2 does not align well with the description in the text, resulting in limited useful information for readers when interpreting the figure.
4.	Please explain Eq. 1 in detail. If it is proposed by yourselves, please clarify the rationality and significance. Or, please cite the original literature at least.
5.	The multiscale augmentation scheme described in Section 3.2 is just a simple down-sampling. I fail to comprehend the reason behind its inclusion as a section.
6.	The details of the encoder and decoder are missing.
7.	The reasons for the three terms in Eq. 7 are the same should be explained and confirmed.
8.	The compared models are insufficient to confirm the usefulness of cross-scale MAE.


**Limitations:**

See “Questions.”

---

> ### Author Rebuttal · Authors · 2023-08-10
>
> Q1&2)
>
> On simplifying the Introduction and clarifying contribution and novelty: As suggested, we will remove Fig. 1 and integrate the information there to Fig. 2. We have also redrawn Fig. 2 with more detailed captions to clarify the contribution. Please refer to Fig. 3 of the rebuttal.
>
> Essentially, compared to SatMAE which is the first paper that applies MAE to extract representations from satellite images with single and fixed scale, Scale-MAE and the proposed Cross-Scale MAE are also based on MAE, but focus on the multi-scale problem. Specifically, Scale-MAE develops the Ground Sample Distance (GSD) position encoding and applies multi de-convolution after decoder to reconstruct images of different scales. Nonetheless, Scale-MAE integrates the scale information into the network via hard coding of known GSD. And, the de-convolution can only result in a specific scale ratio. The proposed Cross-Scale MAE designs the network to learn the information across different scales. With scale augmentation and multi-level contrastive loss between the scale pair and masked patches reconstruction, Cross-Scale MAE can learn informative and consistent representation across different scales without the need of known GSD. Additionally, we leverage xFormer on a single GPU for pre-training efficiency.
>
> While our pipeline may seem straightforward at first glance, it solves three challenging problems. First, given that masking in MAE can be interpreted as a type of augmentation, are there other types of augmentations that can enhance the robustness of representation learning? This paper figures out an intuitive scale augmentation approach that enables scale-invariant representation. Our ablation study (Table 3 of the main text) showed the significant performance gain between using scale augmentation (2nd row) and without it (1st row). Second, given the multiple-scale inputs, most existing pipelines would extract features independently from these inputs without exploiting the cross-scale correlation. In the proposed pipeline, we designed three losses that exploit this cross-scale correlation at two levels, the cross-scale consistency loss at the encoder embed and the cross-scale predictive loss at the decoder embed. In fact, how to use the contrastive loss to calculate the cross-scale consistency loss is a contribution of its own. The ablation study in Table 3 of the main text also showed the significant contribution of each of the cross-scale loss function (Table 3, rows 3-5). Third, the incorporation of xFormers into our pipeline enables the pre-training process to be performed on single GPU, providing a practical pipeline for ViT-based model training.
>
> Q3. On Fig. 2: Thanks for the suggestion. We will remove Fig. 1. We redrew Fig. 2 with more descriptive caption. Please see Fig. 3 in rebuttal.
>
> Q4. On Eq. 1: This is standard positional encoding. We will make sure to add reference and clarify this is part of the background of basic setting, not our proposed work.
>
> Q5.  On the necessity of having Sec. 3.2 as a section: We've dedicated a section to this process due to its pivotal role in our framework. Different from Scale-MAE's complex position encoding, we use a more intuitive approach—scale augmentation with random ratios—to exploit scale information. This enables the network to learn from data. Leveraging scale pairs, we devise multi-stage contrastive learning. Moreover, our framework transforms into a multi-modality fusion tool, accommodating inputs from diverse sensing modalities, like WV and Sentinel 2, or Sentinel 2 and Landsat, without the need for synthesis.
>
> Q6. Our encoder and decoder architecture follows MAE and SatMAE's method, adopting ViT as the backbone. For the encoder, a ViT processes visible patches, embedding them with positional embeddings. Transformer blocks are utilized, excluding masked patches and negating mask tokens. ViT Base employs an embedding dimension of 768 with 12 transformer blocks, each having 12 attention heads. In ViT Large, the embedding dimension is 1024, and the encoder comprises 24 transformer blocks with 16 attention heads.
> The decoder incorporates all tokens, including encoded visible patches and mask tokens, representing missing patches. Mask tokens are shared, learned vectors. Although also using Transformer blocks, the decoder is smaller and shallower than the encoder, enhancing efficiency without compromising meaningful representation learning. During pre-training, the decoder guides the encoder, fostering image reconstruction. Both ViT Base and ViT Large maintain an identical decoder. It boasts an embedding dimension of 512, housing 8 layers with 16 attention heads. Furthermore, Fig. 2 has been updated for enhanced encoder/decoder depiction.Please see Fig. 3 of rebuttal.
>
> Q7. Regarding the terms in Eq. 7: These combined losses comprehensively optimize the proposed Cross-Scale MAE model. The cross-scale consistency loss ($L_{cc}$) ensures uniform and dependable features across diverse scales. The decoder's cross-scale prediction loss ($L_{cp}$) highlights the model's versatile scaling prediction, promoting robustness. Lastly, the reconstruction loss ($L_{re}$) guarantees faithful input-output representation, preserving vital details. Together, these losses offer a holistic optimization strategy, each addressing distinct challenges in multi-scale feature alignment and extraction.
>
> Q8. On compared models being insufficient to showcase the proposed: In the Supplement, we posted more results, including the downstream classification task (Supplement C) and zero-shot transfer learning performance on CoCo (Supplement D). We have since added comparisons with more models and on more downstream tasks. In the rebuttal file, please refer to Table 1 (for comparing segmentation performance with SOTA self-supervised segmentation approaches), and Table 2 (for adding linear classification comparisons to three more models).

---

> > ### Comment · Reviewer_UGSM · 2023-08-13
> > **Final Rate**
> >
> > Thank you for the authors' response. Although they have provided additional explanations and experiments to enrich their work, the proposed methods mostly consist of piecing together existing techniques, and their level of innovation still falls short of the requirements set by NIPS. Therefore, my final decision is to reject the paper.

---

### Official Review · Reviewer_CVyA · 2023-07-07

**Soundness:** 2 fair
**Presentation:** 3 good
**Contribution:** 2 fair
**Rating:** 3
**Confidence:** 4

**Summary:**

The authors propose a Masked-Auto-Encoder approach for training remote sensing data at different scales.  The model is pretrained on the FMoW-RGB and then the representations assessed through KNN classification performance on four other remote sensing datasets.  Comparisons are made to Sat-MAE and Scale-MAE.  Ablation studies are performed to analyze the impact of the different parts of the loss function.  Performance is seen to be higher than Sat-MAE across different amounts of data and on different datasets and generally higher than Scale-MAE on different datasets (performance with different amounts of data not shown).

**Strengths:**

The multi-scale problem in remote sensing is a key challenge that rightly deserves research focus.

The writing is mostly clear, although some sentences or pronoun references could be made clearer in places.

The ablation studies indicate the contribution of the various architectural components.

KNN performance as compared to SAT-MAE is very promising.

A range of datasets at different scales are analyzed.

**Weaknesses:**

Unlike natural images, the scale in most remote sensing tasks is known because of the geoinformation (this is similar to the situation in microscopy where the zoom/scale is known).  One question is whether features across scales is actually the best approach in this regard because the scale from a given sensor is fixed.  A very relevant approach is given by https://github.com/mahmoodlab/HIPT.  Although originally developed for microscopy images, the same notion around handling large image with a known scale size is explored.

In the related works, there is no reference to remote-sensing specific work in this field including approaches like SeCo.  A broad review article is mentioned on, but these other methods are not evaluated for comparison.

Given the comparison to Sat-MAE and Scale-MAE, it would be beneficial to discuss these explicitly in the related works and how the current method differs from them.

It would be beneficial to perform the analysis over multiple runs so the confidence internals and significance of performance difference established.

Figure 3 (performance as a function of data) is only shown in comparison to Sat-MAE which is shown in FIgure 2 to substantially underperform Scale-MAE.  Why not show results from Scale-MAE here as well?

While using KNN classification to compare the representations learned, ultimately this does not measure the main goal of SSL models which is to learn a representation which performs better on the downstream task.  KNN evaluation is one measure, but it is also cirtical to analyze with a (full) frozen decoder as well as a fine-tuned decoder.

Details around training protocol (i.e. learning rate) are missing.  Training on a single GPU is mentioned in 4.3, but it is uncler if this setup is used throughout the work, or only done after the fact as a demonstration that it can be done.

Even though multiple datasets are explored, pretraining is done on a single dataset and then transferred to others.  How does changing which dataset is used impact results?  Furthermore, how does training on multiple datasets with different resolutions work?  In theory this could be the greatest advantage especially if pretraining on low-resolution data (which is often publicly available) could transfer to very high-resolution datasets.

**Questions:**

The caption for Figure 2 could be more descriptive explaining the different components.

It might be more helpful to make Table 5 a chart so the saturation of the performance can be visualized.  It's difficult to tell from the raw numbers if the model has finished training.

The authors reference the varying number of channels used in remote sensing imagery in the introduction, but then do not explore it here.  How can the method be used to handle imagery from sources with a different number of channels.

**Limitations:**

Limitations are not explicitly discussed although possible next steps are briefly mentioned.  No negative societal impacts are expected.

---

> ### Author Rebuttal · Authors · 2023-08-10
>
> Thanks for your meaningful comments. Regarding to your question and comments, our answers are listed as following.
>
> To the weaknesses:
>
> W1. On scale being fixed: We completely agree that remote sensing images usually come with known scale which is fixed for a specific sensing modality. This is why existing practices usually have to pre-train different models to handle images with different scales/resolutions. By introducing scale augmentation as input and cross-scale losses, the Cross-Scale MAE model is able to handle images of various scales using just ONE model. This has made the model "scale-invariant" and largely increased the model generalization capacity. Additionally, even though the paper is motivated by remote sensing applications, the same network can be used to process natural images, where scale information usually is not known apriori. The experimental results shown in Supplement D, where the network is pre-trained on CoCo, validate the effectiveness of the model to scale changes.
> Furthermore, from the ablation study summarized in Tab3 of the main text, we also showed the significant performance gain between using scale augmentation (2nd row) and without it (1st row).
>
> W2. On related works like SeCo: We will include references to more related works like SeCo in addition to the broad review article. We also added comparisons to more methods to Tab3 in Supplement C. The new table (Tab2 of rebuttal) shows Cross-Scale MAE outperforms SeCo and Scale-MAE in the linear classification task.
>
> W3. On more discussions to SatMAE and Scale-MAE: We will add the following details. SatMAE is the first paper that applies MAE to extract representations from satellite images but with single and fixed scale. Scale-MAE and the proposed Cross-Scale MAE are also based on MAE, but focus on the multi-scale problem. Specifically, Scale-MAE develops the Ground Sample Distance (GSD) position encoding and applies multi de-convolution after decoder to reconstruct images of different scales. Nonetheless, Scale-MAE integrates the scale information into the network via hard coding of known GSD. And, the de-convolution can only result in a specific scale ratio. The proposed Cross-Scale MAE designs the network to learn the information across different scales. With scale augmentation and multi-level contrastive loss between the scale pair and masked patches reconstruction, it can learn informative and consistent representation across different scales without the need of known GSD.
>
> W4. On the necessity of multiple runs: When employing the ViT backbone, we noticed that displaying discrepancies across multiple training runs isn't customary, echoing practices in related works like SatMAE and Scale-MAE. This stability is influenced by consistent loss convergence and the considerable compute resources needed for training, discouraging multiple runs of the same model setup. Our development process encompassed training multiple models, revealing a consistent convergence pattern across them. Hence, we opted against training multiple runs for the same final model configuration. However, testing runs underwent scrutiny; the results are available in the Supplement (Figs. 3, 5, 8). These graphs stem from 25 runs per testing set sample, with each run involving varied cropping and scaling. In every run, uniform image scale and masking are employed for fair comparison.
>
> W5. On adding Scale-MAE performance to Fig. 3: The Scale-MAE model was not open source until recently. We are thus able to add its performance curve to Fig. 3. Please check Fig.2 of the rebuttal.
>
> W6. On more downstream tasks: We completely agree that having kNN evaluation is insufficient. Due to the usage of cross-scale contrastive loss at the encoder and a cross-scale predictive loss at the decoder, the extracted representation would possess both discriminative and representative characteristics. Hence, downstream tasks should include both classification and segmentation performance. We have since conducted additional experiments on segmentation that further demonstrate consistent improvements over SOTA self-supervised segmentations. The results are shown in Tab1 of the rebuttal.
>
> W7. On details of training protocol. The training details, including hyperparameter choices, can be found in Supplement C (2nd paragraph). We should have referred to it in the main text. It's important to note that all other hyperparameters remained consistent with those utilized for the final models presented in the MAE and SatMAE papers we built upon. Regarding the training GPU setup, we confirm that the models were indeed trained on a single GPU throughout the entirety of our work.
>
> W8. On using multiple datasets with different resolutions to train: The fMoW dataset is indeed a multi-scale dataset with the Ground Sample Distance (GSD) varying from 0.3m-3m. Also, during training, we assume the actual GSD is unknown and apply scale augmentation to synthesize image pairs with different scales. So the trained model can be used to handle input images with a wide range of scale variation - it can handle both single-scale and multi-scale dataset without the need of the ground truth GSD. Besides fMoW, we also pre-trained the model on CoCo to evaluate its effectiveness on natural images. The results were provided in Supplement D.
>
> Q1. On Fig2. We redrew Fig2 with more descriptive caption. Please see Fig3 in rebuttal.
>
> Q2. On Tab5: We converted Tab5 to a line chart. Please see Fig1 in rebuttal.
>
> Q3. On imagery with different channels: This is indeed a challenging problem and has been dealt with in areas like domain adaptation and channel harmonization, where different channels are preprocessed to map to a reference set of channels. Cross-Scale MAE cannot fundamentally solve this problem. For now, we pick the subset of bands shared by different sensing modalities, i.e., RGB and near-infrared.

---

### Author Rebuttal · Authors · 2023-08-10

We are appreciate all reviewers' significant comments.

Initially, we wish to elucidate that our supplementary material, accompanying the main paper, encompasses a wealth of substantial experimental outcomes. This includes the visualization of multiscale representation benefits, downstream task analyses, zero-shot transfer learning demonstrations, and an evaluation of xFormers. We believe that several concerns raised by reviewers can be addressed effectively through the supplementary material.

Furthermore, to comprehensively address reviewer queries, we've curated a rebuttal document. This document integrates additional experiment results as requested by reviewers and incorporates refined figures in accordance with their suggestions.

We trust that our responses and explanations address your inquiries and uncertainties. Feel free to inquire further about our explanations.

---

### Decision · Program_Chairs · 2023-09-21

**Decision:**

Accept (poster)

**Comment:**

The reviewers strongly disagree about this paper. Having read the paper, reviews and rebuttal, the AC find themself agreeing with the most positive reviewers that - while building on existing idea around SSL, MAE and scale augmentation - the paper reports strong results, of interest to the community, with a simple well explained and well presented method, and therefore is happy to recommend the paper for acceptance.

The authors should incorporate additional results from the sup. mat. and the rebuttal in the final version of the paper.